 **eLIFE**

# An unmet actin requirement explains the mitotic inhibition of clathrin-mediated endocytosis

**Satdip Kaur[1], Andrew B Fielding[2], Gisela Gassner[2], Nicholas J Carter[1], Stephen J Royle[1,2]\***

[1]Division of Biomedical Cell Biology, Warwick Medical School, University of Warwick, Coventry, United Kingdom; [2]Department of Cellular and Molecular Physiology, Institute of Translational Medicine, University of Liverpool, Liverpool, United Kingdom

**Abstract** Clathrin-mediated endocytosis (CME) is the major internalisation route for many different receptor types in mammalian cells. CME is shut down during early mitosis, but the mechanism of this inhibition is unclear. In this study, we show that the mitotic shutdown is due to an unmet requirement for actin in CME. In mitotic cells, membrane tension is increased and this invokes a requirement for the actin cytoskeleton to assist the CME machinery to overcome the increased load. However, the actin cytoskeleton is engaged in the formation of a rigid cortex in mitotic cells and is therefore unavailable for deployment. We demonstrate that CME can be 'restarted' in mitotic cells despite high membrane tension, by allowing actin to engage in endocytosis. Mitotic phosphorylation of endocytic proteins is maintained in mitotic cells with restored CME, indicating that direct phosphorylation of the CME machinery does not account for shutdown.

**\*For correspondence:** s.j.royle@warwick.ac.uk

**Competing interests:** The authors declare that no competing interests exist.

**Reviewing editor**: Pascale Cossart, Institut Pasteur, France

## Introduction

Clathrin-mediated endocytosis (CME), the major route of internalisation for transmembrane proteins into mammalian cells, controls many cellular processes from signalling and growth, to cell motility and organelle identity (*McMahon and Boucrot, 2011*). It has been known for decades that CME, like many other cellular processes, is inhibited during the early stages of mitosis (*Fawcett, 1965*; *Berlin and Oliver, 1980*; *Warren, 1993*; *Fielding et al., 2012*; *Fielding and Royle, 2013*). However, the mechanism that underlies this mitotic inhibition is unclear (*Fielding and Royle, 2013*).

During CME, the cargo that is destined for internalisation is clustered into pits via interaction with adaptor proteins, which in turn, recruit clathrin triskelia. The forming clathrin-coated pit is then invaginated further and eventually pinched off in a process dependent on the GTPase dynamin. The actin cytoskeleton may assist in these latter stages of vesicle formation (*Robertson et al., 2009*).

CME is shut down shortly after prophase entry and resumes in late anaphase where it is required for membrane dynamics in cytokinesis (*Berlin and Oliver, 1980*; *Schweitzer et al., 2005*). In early mitotic cells, shallow clathrin-coated pits are seen at the plasma membrane suggesting that invagination and subsequent steps of CME are inhibited (*Pypaert et al., 1987*). Two main mechanisms have been proposed. First, direct mitotic phosphorylation of the CME machinery decreases its activity (*Pypaert et al., 1991*; *Chen et al., 1999*; *He et al., 2003*). In support of this, numerous endocytic proteins are phosphorylated during mitosis (*Dephoure et al., 2008*), but it is not clear what effect, if any, these modifications have on CME. Second, that the increased membrane tension of mitotic cells prevents invagination during CME (*Raucher and Sheetz, 1999*). There is good evidence that the membrane

**eLife digest** The plasma membrane that surrounds a cell acts as a protective barrier that regulates what can enter or exit the cell. However, large molecules and other 'cargo' can get into a cell in a variety of ways. One of these routes—known as clathrin-mediated endocytosis—involves a receptor on the outside of the membrane grabbing hold of the cargo while a protein called clathrin forms a 'pit' beneath the receptor. This pit becomes deeper and deeper until the cargo is completely surrounded by clathrin-lined membrane and is brought inside the cell.

This process has been studied over the past 50 years, and it is known that clathrin-mediated endocytosis is turned off when a cell begins to divide to produce new cells, and then turned back on when cell division has come to an end. However, there are competing theories as to exactly why this process stops when cell division starts.

Now, Kaur et al. have investigated these theories by looking at the role that another protein, called actin, plays in turning off clathrin-mediated endocytosis. Actin is a molecule that forms a sort of scaffolding within the cell (called the cytoskeleton), and it also guides the movement of molecules and larger structures within the cell. Further, when the cell membrane is being stretched, the actin cytoskeleton can assist the clathrin-mediated endocytosis machinery to pull cargo into the cell.

So why doesn't actin help with endocytosis during cell division? The answer, Kaur et al. suggest, is that all the actin in the cell is needed by the cytoskeleton during cell division, so there is no actin available to perform other tasks such as clathrin-mediated endocytosis. Further experiments demonstrated that this form of endocytosis can be 'restarted' in dividing cells by treating the cells in a way that frees up some additional actin. The work of Kaur et al. also ruled out the theory that chemical changes to the endocytosis machinery disabled it during cell division. These findings have implications for the delivery of drugs, via endocytosis, to the rapidly dividing cells that are involved in diseases such as cancer.

tension is increased in early mitosis (*Stewart et al., 2011a*) in a process driven by osmotic changes (*Habela and Sontheimer, 2007*; *Stewart et al., 2011b*). The precise mechanism for shutdown is yet to be determined.

Despite a clear role in yeast, the actin cytoskeleton does not have an obligatory role in CME in human cells (*Engqvist-Goldstein and Drubin, 2003*; *Robertson et al., 2009*; *Traub, 2011*; *Anitei and Hoflack, 2012*). Light and electron microscopy studies clearly show that it is recruited to a subset of CME events in human cells (*Merrifield et al., 2002, 2005*; *Collins et al., 2011*; *Taylor et al., 2011, 2012*) and that it is important for internalisation of pathogens such as Listeria (*Bonazzi et al., 2011*). Interestingly, the requirement for actin in CME may be linked to membrane tension. Recent work has shown that in cells where membrane tension is increased, there is an increased requirement for actin (*Aghamohammadzadeh and Ayscough, 2009*; *Boulant et al., 2011*; *Stachowiak et al., 2013*). During mitosis the plasma membrane tension is increased as a result of osmotic changes (*Raucher and Sheetz, 1999*; *Stewart et al., 2011a*), so why is the actin cytoskeleton not deployed to assist CME as occurs in interphase cells?

We set out to discover the mechanism for mitotic shutdown of CME. We found two methods to 'restart' CME in mitotic cells, which effectively rule out direct mitotic phosphorylation of endocytic proteins as a mechanism for the mitotic shutdown of CME. Instead we propose that the actin machinery, which is engaged in assembling a stiff cortex in mitotic cells, is unavailable to rescue the inhibition caused by the increased membrane tension. This unmet requirement for actin in CME explains why endocytosis is inhibited in early mitosis.

## Results and discussion

### Membrane tension is elevated in mitotic cells

Previous reports indicated that due to osmotic changes in mitosis, the plasma membrane tension is increased (*Raucher and Sheetz, 1999*; *Stewart et al., 2011b*). We began by verifying this observation in HeLa cells. To measure apparent membrane tension, a polystyrene bead coated with concanavalin A was held in an optical trap, attached to the plasma membrane, and then pulled away from the cell

surface to form a membrane tether (*Dai and Sheetz, 1995*; *Figure 1A,B*). The displacement of the bead from the trap centre is the tether force and this is proportional to the plasma membrane tension (*Figure 1A*). We found that the tether force was increased from 19.2 ± 1.5 pN in interphase cells to 27 ± 2.8 pN in mitotic cells at metaphase (mean ± SEM, p=0.027, *Figure 1C*). These measurements are in good agreement with previous measurements in other cell types (*Raucher and Sheetz, 1999*). Tether force is related to the force required to invaginate a clathrin-coated vesicle. This means that, in mitotic cells, comparatively more energy is required from the CME machinery as it tries to overcome an increased load. The shortfall in this energy may explain mitotic shutdown of CME and the question is: what is the molecular basis of this shortfall?

## Differences between clathrin-coated structures in mitosis vs interphase

Clathrin-coated structures (CCSs, pits, and vesicles) are present in mitotic cells with the pits being arrested at the cell surface in a shallow state (*Pypaert et al., 1987*). Fluorescent transferrin binding to abundant transferrin receptors on the plasma membrane can be seen in these pits but the uptake of transferrin is prevented (*Figure 2A*; *Fielding et al., 2012*). Our hypothesis was that differences in the proteomes of CCSs purified from interphase or mitotic cells would explain why clathrin-coated pits are arrested at the surface. For example, components of the CME machinery that are regulated might be

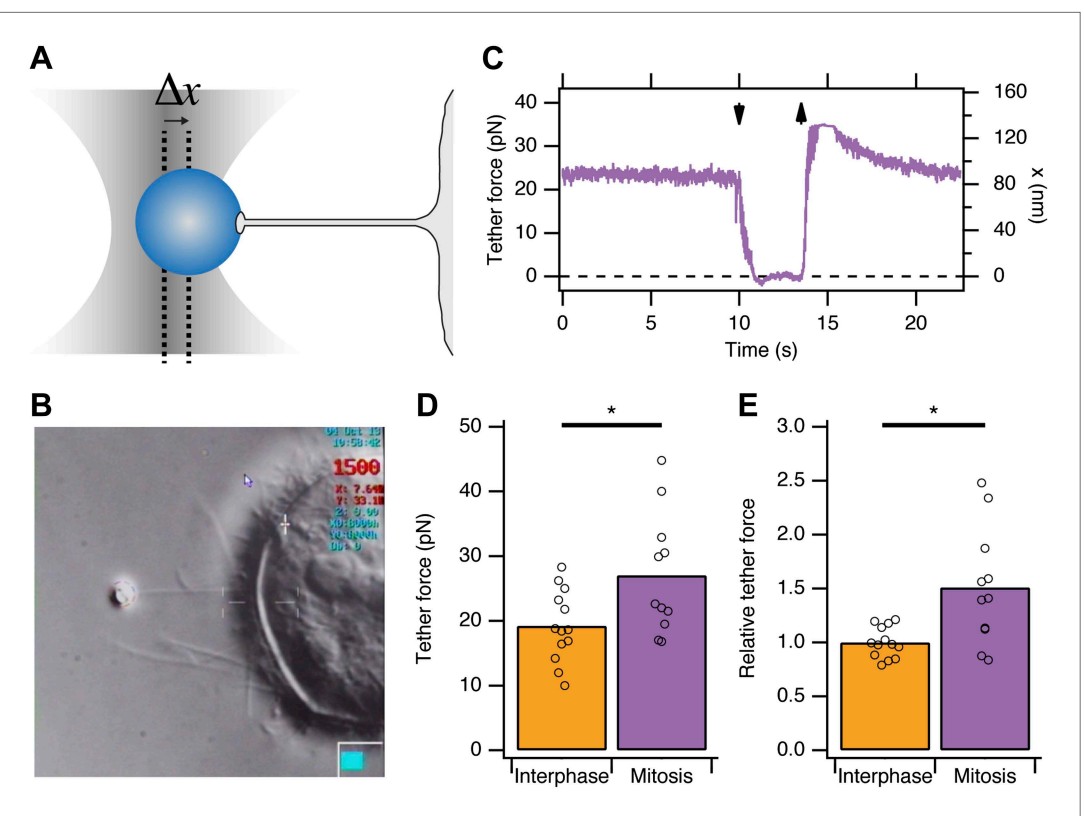

**Figure 1**. Membrane tension in mitotic cells is elevated. (**A**) Schematic diagram to show the principle of using tether force to measure membrane tension. Displacement of the bead from the trap centre (Δx) can be converted to a tether force measurement (pN) as the stiffness of the trap is known. This tether force is proportional to the membrane tension. (**B**) Still video image of a membrane tension measurement in a mitotic cell. Note the thin tether is formed by moving the cell away from the bead held in the trap. (**C**) A typical trace showing bead displacement from the trap centre and the tether force. The cell was moved towards the bead (down arrow) to eliminate tension and give a zero value. The cell is then moved away (up arrow) and tether force is re-established. (**D**) Summary plot of tether force measurements from HeLa cells (dots) in interphase (orange) or at metaphase (purple). The mean is shown by a bar. (**E**) Plot of tether force relative to the average interphase measurement from the same experiment. $N_{cell}$ = 13 or 11, $N_{exp}$ = 5. *p<0.05.

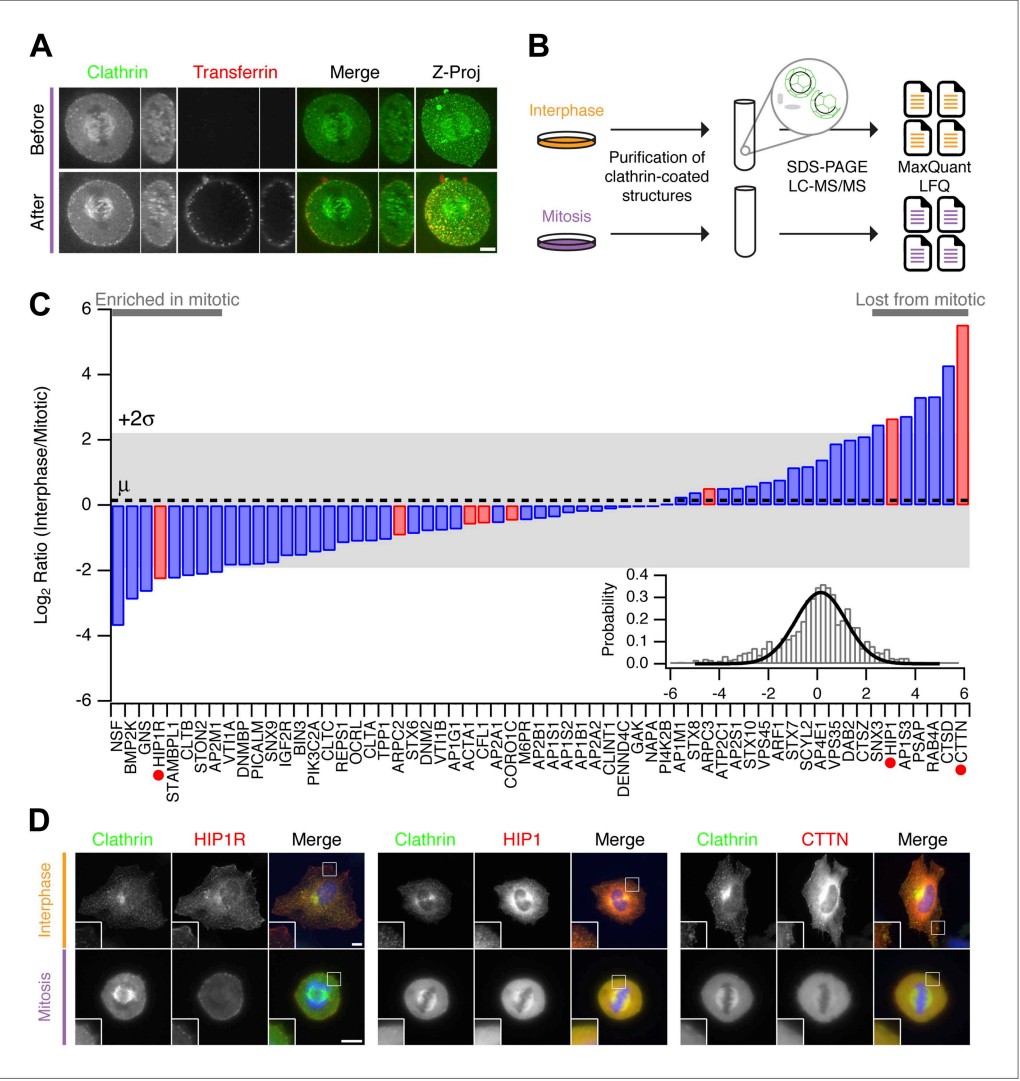

**Figure 2**. Comparative proteomics of fractions containing clathrin-coated membranes purified from interphase and mitotic HeLa cells. (**A**) Confocal micrographs of a HeLa cell expressing GFP-tagged clathrin light chain a (clathrin, green). The same cell is shown before and after application of transferrin-Alexa568 (Tf, red). A single XY section is shown together with a YZ view through the cell centre (right). Far right, 3D projection of the whole confocal series. (**B**) Schematic diagram of the purification and proteomic analysis of clathrin-coated structures purified from cells in interphase or metaphase. (**C**) Bar chart to show the comparative interphase/mitotic abundance for CCV proteins from the search list. The interphase or mitotic LFQ value for each protein derived from four separate experiments were compared. Red bars show proteins related to the actin cytoskeleton. Red circles indicate proteins verified in **D**. Inset: histogram to show the frequency of abundances for all proteins in the analysis. A single Gaussian function was fitted to the data, the mean and variance of which is shown in the main bar chart. Dotted line shows the average comparative abundance and the shaded area represents ± 2 standard deviations. (**D**) Representative fluorescence micrographs to show the colocalisation of GFP-tagged clathrin light chain a (clathrin, green) in interphase or mitosis, with HIP1R-tDimer-RFP, tdTomato-HIP1 or mCherry-cortactin (red). Scale bar, 10 μm.

present in interphase CCSs but could be less abundant in mitotic CCSs. To identify such differences, we prepared fractions enriched in CCSs from interphase or mitotic HeLa cells, according to established methods (**Borner et al., 2006**, **2012**). These fractions were analysed by mass spectrometry and label-free quantitation (**Figure 2B**). Over four independent experiments, we compared the relative abundance of 1253 proteins, only a subset of which are confirmed CCS proteins (**Borner et al., 2006**, **2012**). The list of all abundances was used to characterise the variance of the dataset and identify outliers ('Materials and methods'). We found that cortactin was the protein most consistently reduced in mitotic

CCSs compared to interphase (LFQ intensity ratio = 46.2). Cortactin is an activator of Arp2/3-dependent actin polymerisation. A list of bona fide CCS proteins (*Borner et al., 2012*) was therefore supplemented with components of the actin cytoskeleton that are recruited to CCSs (*Taylor et al., 2011*). The relative abundance of these proteins is shown in *Figure 2C*. These data show that most of the core CME machinery is not altered significantly between CCS-containing fractions from interphase and mitotic samples. Consistent with previous results (*Chetrit et al., 2011*; *Kozik et al., 2013*), Dab2 was less abundant and PICALM more abundant in mitotic fractions. HIP1 and HIP1R appeared to be differentially regulated. These two proteins link the clathrin machinery with the actin cytoskeleton and are regulated by binding clathrin light chain (*Le Clainche et al., 2007*; *Wilbur et al., 2008*). The accumulation of HIP1R and the absence of cortactin in mitotic fractions were interesting, given that these two proteins have been shown previously to be coupled functionally (*Le Clainche et al., 2007*). Other interesting differences, to be explored in the future, included the accumulation of NSF (N-Ethylmaleimide-Sensitive Factor) in mitotic CCSs (LFQ intensity ratio = 0.077).

We next verified the comparative proteomics results by visualising the co-localisation of clathrin with HIP1R, HIP1, and cortactin in interphase and mitotic cells. All three proteins were colocalised at least partially with clathrin in interphase CCSs. In mitotic cells, HIP1R was found tightly associated with clathrin-coated pits at the cell surface (*Figure 2D*). By contrast, HIP1 and cortactin were diffusely localised in mitotic cells and were lost from CCSs (*Figure 2D*). These results indicate that the actin cytoskeleton is important for the mitotic shutdown of CME.

Taken together, this suggested the following mechanism: it is predicted that actin is required to assist CME in mitotic cells due to the increase in membrane tension (*Aghamohammadzadeh and Ayscough, 2009*; *Boulant et al., 2011*). If the actin cytoskeleton is engaged in the formation of a stiff mitotic cortex (*Bray and White, 1988*; *Cramer and Mitchison, 1997*; *Stewart et al., 2011a*), perhaps it is unavailable to assist CME in mitotic cells?

## 'Restarting' CME in mitotic cells: Ect2 depletion

To test this idea, we developed two strategies to 'restart' CME in mitotic cells. Both methods work by allowing the actin cytoskeleton to engage in CME in mitotic cells. Our first strategy was to deplete Ect2, a guanine nucleotide exchange factor effector for RhoA. Mitotic cells depleted of Ect2 showed similar CME to control cells in interphase (*Figure 3A*). Quantification of transferrin uptake demonstrated that with either of two independent siRNAs against Ect2, the mitotic shutdown of CME was reversed (*Figure 3A–C*). Furthermore, this reversal could itself be blocked by the expression of siRNA-resistant GFP-Ect2 constructs (*Figure 3—figure supplement 1*). These experiments show that restoration of CME following Ect2 RNAi is specifically due to loss of Ect2 protein and not due to an off-target effect.

It was described previously that Ect2-depletion inhibits the formation of the stiff actin cortex in mitotic cells (*Matthews et al., 2012*). We therefore assessed actin availability to see if this was altered by Ect2 depletion. In mitotic cells depleted of Ect2, the ratio of F-actin fluorescence at the cell cortex vs the cytoplasm was reduced compared with control RNAi cells (*Figure 3D,E*). Analysis of cytosolic G-actin, visualised by DNase I staining (*Cramer et al., 2002*) revealed small but significant increases in the level of G-actin in mitotic cells depleted of Ect2 compared with control RNAi cells (*Figure 3F*). These observations suggest that Ect2 depletion increases the availability of actin because the cortical F-actin is not efficiently remodelled into a cortex.

We next measured the membrane tension of mitotic cells following depletion of Ect2. The relative tether force in mitotic cells depleted of Ect2 was not significantly different from control RNAi mitotic cells (1.02 ± 0.15, mean ± SEM, p=0.921, Student's *t* test) (*Figure 3G*). This ruled out the possibility that actin cytoskeleton changes in Ect2-depletion indirectly decrease membrane tension and therefore restore CME in mitotic cells. The measurement shows that membrane tension remains high in mitotic Ect2-depleted cells.

Is the restored CME in Ect2-depleted mitotic cells actin-dependent? To test this possibility, we treated cells with latrunculin B (1 µM), a toxin that promotes actin disassembly. We found that restored CME in Ect2-depleted mitotic cells was sensitive to latrunculin B, whereas CME in interphase was not (*Figure 3H*). Note that there was no evidence of restarting CME in control mitotic cells treated with latrunculin B (*Figure 3H*). This argues against the possibility that the remodelled actin cortex in mitotic cells directly inhibits CME, as its destruction by the drug did not result in endocytosis. These experiments show that restored CME in mitotic cells is different to CME in interphase cells: it is strictly actin-dependent.

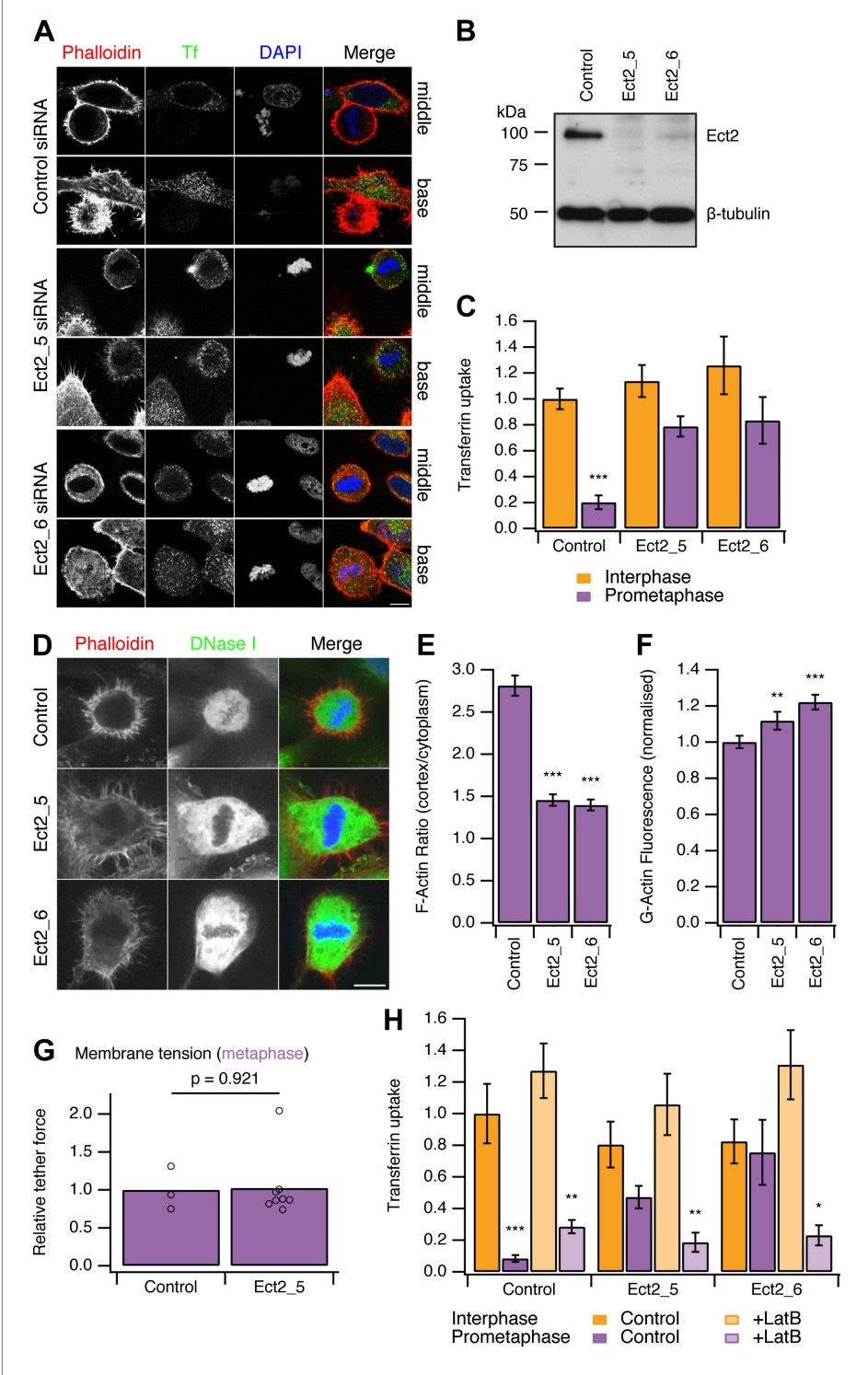

**Figure 3**. Restoring CME in mitotic cells by depletion of Ect2. (**A**) Representative confocal micrographs to show transferrin uptake (Tf, green), actin organisation (phalloidin, red) and DNA (blue) in control or Ect2-depleted cells. For each condition, a section though the middle or base of the mitotic cell with a neighbouring interphase cell is shown. Cells were transfected with control siRNA or one of two Ect2 siRNAs (Ect2_5 or Ect2_6). Scale bars, 10 μm. (**B**) Representative western blot to show the amount of Ect2 remaining after RNAi. Blots were probed for Ect2 and β-tubulin as a loading control. (**C**) Bar chart to show normalised transferrin uptake in interphase and mitotic
*Figure 3. Continued on next page*

*Figure 3. Continued*

(prometaphase) cells as quantified by confocal microscopy. N<sub>cell</sub> = 5–8. (**D**) Representative confocal micrographs to show the distribution of F-actin (Phalloidin, red), G-actin (DNase I, green) and DNA (blue) in mitotic HeLa cells. The cells were transfected with control (GL2) or Ect2 (Ect2_5 or Ect2_6) siRNA. Scale bar, 10 μm. (**E** and **F**) Bar charts to summarise the quantification of F-actin ratio at the cell cortex vs the cytoplasm (**E**) or the amount of G-actin in the cytoplasm (**F**). $N_{cell}$ = 40, $N_{exp}$ = 3. (**G**) Summary of membrane tension measurements on mitotic HeLa cells transfected with control (GL2) or Ect2 (Ect2_5) siRNA. Relative tether force is shown for individual cells (dots) and the bar indicates the mean. (**H**) Bar chart to show normalised transferrin uptake in interphase and mitotic (prometaphase) cells. Note the inhibition by latrunculin B (LatB, 1 μM 30 min) of restored CME in mitotic cells depleted of Ect2. $N_{cell}$ = 7–14. All bars show mean ± SEM, *p<0.05; **p<0.01; ***p<0.001. One-way ANOVA with Tukey's post-hoc test, comparison to control RNAi, interphase.

The following figure supplements are available for figure 3:

**Figure supplement 1**. Ect2 RNAi rescue experiment to show that restoration of CME in Ect2-depleted cells is due to loss of Ect2.

Together our data demonstrate that the actin cytoskeleton is required for CME in the face of increased membrane tension during mitosis.

## 'Restarting' CME in mitotic cells: Rap1 activity

Our second strategy to 'restart' CME in mitotic cells was to use a constitutively-active form of the small GTPase Rap1, Rap1(Q63E) (*Dao et al., 2009*). Rap1 is an activator of β1-, β2- or β3-containing integrins that connect to the actin cytoskeleton (*Kim et al., 2011*). At mitosis onset, Rap1 is normally inactivated, allowing the disassembly of focal adhesions and actin stress fibres, resulting in cell rounding (*Dao et al., 2009*). Expression of Rap1(Q63E), but not the inactive mutant Rap1(S17A), causes mitotic cells to remain flat and unable to form a rounded actin cortex (*Figure 4A*; *Dao et al., 2009*). Measurement of the ratio of F-actin fluorescence at the cell cortex vs the cytoplasm showed a decrease in Rap1(Q63E)-expressing cells relative to controls (*Figure 4B*). The inhibition of cortex formation resulted in ~20% more G-actin to be available in mitotic cells expressing Rap1(Q63E) compared with control mitotic cells (*Figure 4C*). Moreover, membrane tension remained high in mitotic cells expressing Rap1(Q63E) (*Figure 4D*). The relative tether force was 1.15 ± 0.08 (mean ± SEM, p=0.21, Student's *t* test). These results indicate that, as with Ect2 depletion, the membrane tension remains high but the prevention of F-actin remodelling in mitosis has resulted in more actin becoming available to assist CME.

To test if this manipulation also 'restarts' CME in mitotic cells, we next measured uptake of fluorescent transferrin. We found that mitotic cells expressing Rap1(Q63E) exhibited good transferrin uptake and this was comparable to the amount seen in interphase cells (*Figure 4E,F*).

We next investigated the involvement of the actin cytoskeleton in restored CME in Rap1(Q63E)-expressing cells. To do this, we depleted cortactin using RNAi (*Figure 4E,F*). Depletion of cortactin blocked the restored CME in Rap1(Q63E)-expressing cells (*Figure 4E,F*). This is in contrast to interphase cells where depletion of cortactin had no effect on CME as measured by transferrin uptake (*Figure 4E,F*). The cortactin-dependence of restored CME in Rap1(Q63E)-expressing cells was not due to an off-target effect of RNAi. First, it could be demonstrated with two different siRNAs (*Figure 4E,F*). Second, the effect could be rescued by expression of an siRNA resistant form of cortactin in Rap1(Q63E)-expressing cells (*Figure 4—figure supplement 1*). This demonstrates that 'restarted' CME in mitotic cells is different from CME in interphase: it is dependent on cortactin, a protein that was lost from mitotic CCSs in our proteomic analysis (*Figure 2*). This indicated that using Rap1(Q63E) expression to restart CME in mitotic cells and then inhibiting the restored CME using cortactin depletion, is analogous to the approach above using Ect2 RNAi and latrunculin B. Inhibition of actin function by latrunculin B or cortactin RNAi demonstrate that CME in mitotic cells requires actin and that Ect2 RNAi or Rap1(Q63E)-expression permit the utilisation of actin to restore CME. Both strategies to restart mitotic CME support a model where an unmet requirement for actin in CME due to increased membrane tension, causes shutdown.

Does 'restarted' CME in mitotic cells represent global restoration of CME, or is it specific to CME of the transferrin receptor? To explore this question, we tested for restoration of internalisation of the three main endocytic motifs YXXΦ, [DE]XXXL[LI], and FXNPXY using the CD8 reporter system

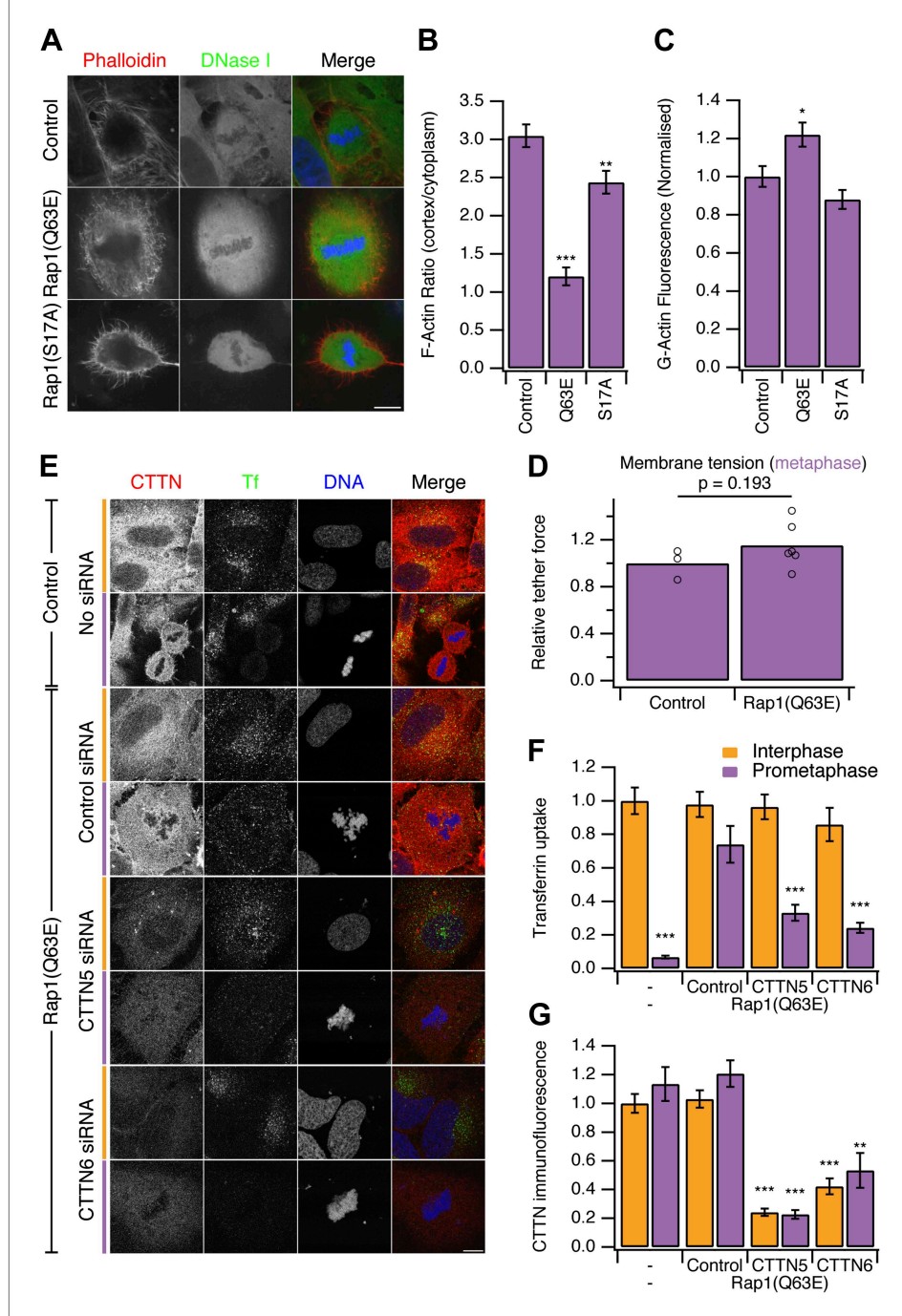

**Figure 4**. Restoring CME in mitotic cells by expression of Rap1(Q63E). (**A**) Representative confocal micrographs to show the distribution of F-actin (Phalloidin, red), G-actin (DNase I, green), and DNA (blue) in mitotic HeLa cells. The cells either expressed no protein, the constitutively active Rap1(Q63E) or the inactive Rap1(S17A). Scale bar, 10 μm. (**B** and **C**) Bar charts to summarise the quantification of F-actin ratio at the cell cortex vs the cytoplasm (**B**) or the amount of G-actin in the cytoplasm (**C**). $N_{cell}$ = 18. (**D**) Summary of membrane tension measurements on mitotic HeLa cells expressing Rap1(Q63E) compared to control. Relative tether force is shown for individual cells (dots) and the bar indicates the mean. (**E**) Representative confocal micrographs of interphase (orange) or mitotic (purple) HeLa cells to show transferrin uptake (Tf, green), cortactin immunofluorescence (CTTN, red) and DNA (blue). The cells were transfected as indicated to express Rap1(Q63E) and with control or either one of two CTTN siRNAs (CTTN_5 or CTTN_6). Scale bar, 10 μm. (**F** and **G**) Bar charts to show normalised transferrin uptake (**F**) or CTTN immunofluorescence

*Figure 4. Continued on next page*

*Figure 4. Continued*

(**G**) for each condition as indicated. $N_{cell}$ = 10–29. All bar charts in the figure show mean ± SEM. All bars show mean ± SEM, *p<0.05; **p<0.01, ***p<0.001. One-way ANOVA with Tukey's post-hoc test, comparison to control (**B** and **C**) or normal HeLa, interphase (**F** and **G**).

The following figure supplements are available for figure 4:

**Figure supplement 1**. Cortactin RNAi prevents the restoration of CME by Rap1(Q63E) expression, and this effect is due to loss of cortactin.

(*Kozik et al., 2010*; *Fielding et al., 2012*). Confocal imaging of the steady-state distribution of CD8 chimeras in control mitotic cells showed no intracellular puncta and all CD8 reporters localised at the plasma membrane (*Figure 5*). In mitotic cells expressing Rap1(Q63E), there were numerous intracellular puncta consistent with CME of CD8 chimeras possessing any of the three main endocytic motifs (*Figure 5A*). Note that this assay of endocytosis does not involve temperature shifts, chemical synchronisation or serum starvation, which have been proposed to affect CME (*Tacheva-Grigorova et al., 2013*). Moreover, endocytosis of CD8 chimeras as measured by antibody labelling on live cells using a fluorophore-conjugated anti-CD8 antibody confirmed that uptake of CD8 chimeras with endocytic motifs occurred in Rap1(Q63E) expressing mitotic cells (*Figure 5B*).

## Mitotic phosphorylation of endocytic proteins does not account for mitotic shutdown of CME

An alternative explanation for CME shutdown in mitosis is the phosphorylation of endocytic proteins by mitotic kinases (*Pypaert et al., 1991*; *Chen et al., 1999*; *Fielding and Royle, 2013*). Our observations, that mitotic cells are competent for CME if the actin cytoskeleton is made available, runs counter to this hypothesis. We first tested if mitotic phosphorylation of endocytic proteins was maintained in cells with restored CME. We analysed the phosphorylation status of Dab2 by western blotting. Dab2 is a clathrin adaptor for cargo proteins with an NPXY endocytic motif (*Mishra et al., 2002*) and it is known to be phosphorylated in mitosis (*He et al., 2003*; *Chetrit et al., 2011*). *Figure 6A* shows that Dab2 is phosphorylated in mitosis and that this phosphorylation can be reversed by brief treatment of mitotic cells with the Cdk1 inhibitor flavopiridol (5 µM, 10 min) (*Losiewicz et al., 1994*). An identical situation was seen in lysates from cells expressing GFP-Rap1(Q63E) (*Figure 6A*). This suggests that CME of proteins with NPXY motifs can occur (*Figure 5*) despite maintained mitotic phosphorylation of Dab2 by Cdk1. Secondly, we tested whether inhibition of Cdk1 was sufficient to restart CME. Such restarting would be expected if mitotic phosphorylation inhibited the CME machinery directly. To do this, CME was measured using transferrin uptake and flow cytometry. The brief incubation of mitotic cells with flavopiridol (5 or 10 µM) did not restart CME. Together, these results indicate that the direct mitotic phosphorylation of endocytic proteins cannot account for mitotic shut down of CME. It remains possible that phosphorylation of endocytic proteins may have a more minor, modulatory role in mitotic shutdown of CME, but we found no evidence that phosphorylation of the CME machinery renders it inactive. Moreover, it should be noted that cyclin B1-Cdk1 activity orchestrates mitosis and therefore drives the changes that result in the inhibition of CME at a higher level.

We set out to describe the mechanism by which cells shut down CME during early mitosis. Membrane tension is elevated in mitotic cells and presents a challenge to the CME machinery: it now requires the assistance of actin to overcome this tension and internalise receptors. A proteomic survey revealed a comparative loss of cortactin from mitotic CCSs, implicating the actin cytoskeleton as being key to the inhibition. We showed that mitotic shutdown of CME can be cancelled either by the loss of Ect2 or by constitutive activation of Rap1, two manipulations that increase actin availability by affecting the formation of a rounded actin cortex in mitotic cells. Restored CME under these conditions was uniquely sensitive to latrunculin B-treatment or cortactin-depletion, respectively. This agrees with the idea that the actin cytoskeleton is required for CME in mitotic cells, but it is normally unavailable due to the formation of the cortex. The unmet actin requirement for CME in mitosis is shown schematically in *Figure 7*.

The mechanism for mitotic shut down of CME can therefore be thought of as a 'two layer' inhibition. The first layer of inhibition comes from an increase in membrane tension, mainly due to osmotic

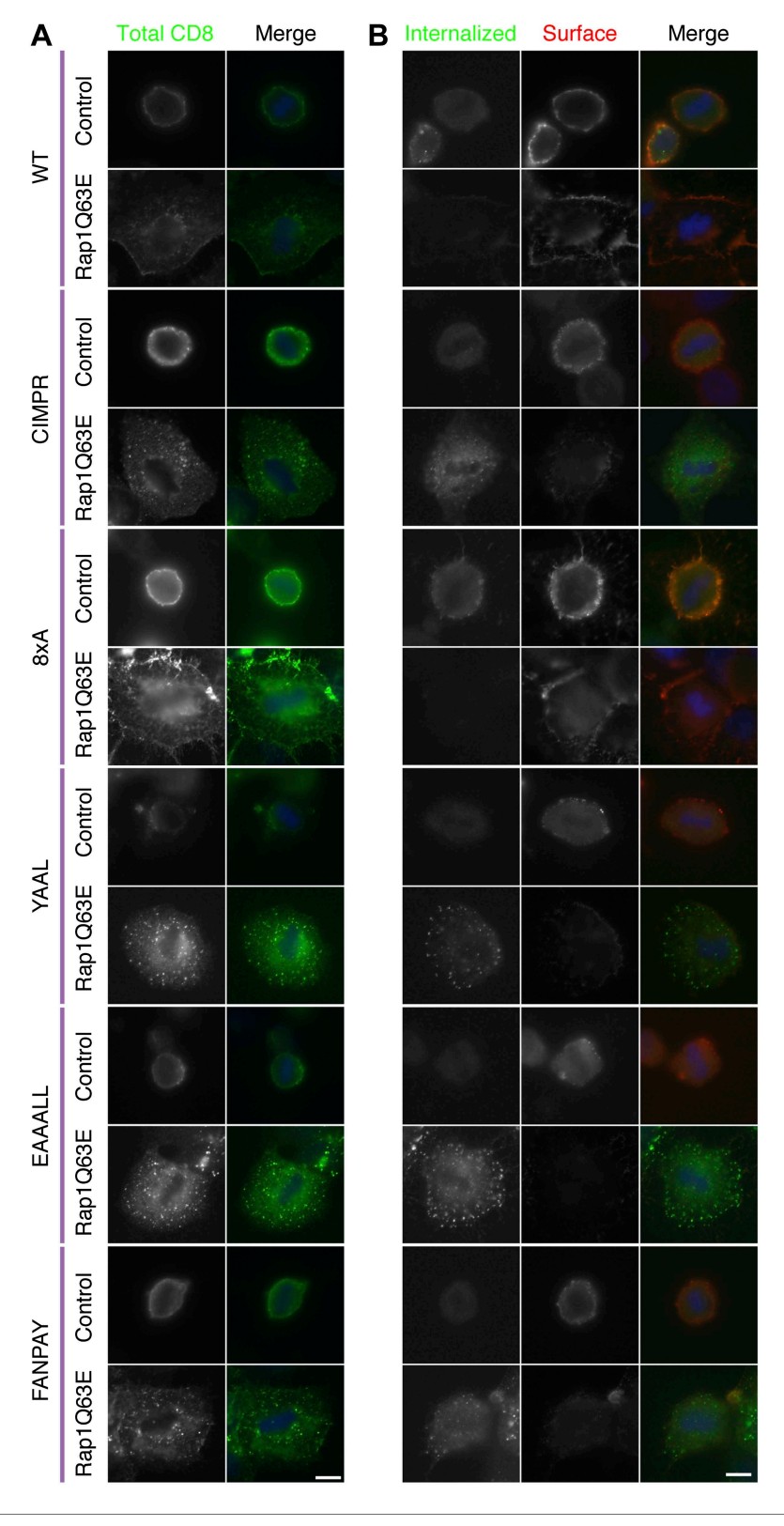

**Figure 5**. Internalisation of diverse cargo is restarted in mitotic Rap1(Q63E)-expressing cells. (**A**) Representative confocal images of control and Rap1(Q63E)-expressing HeLa cells in mitosis that express the indicated CD8 chimera. The cells were fixed and stained for total CD8 to show the steady-state distribution of CD8 chimeras (green; DNA, blue).
*Figure 5. Continued on next page*

*Figure 5. Continued*

(**B**) Representative fluorescence micrographs of anti-CD8 antibody uptake experiments. Control and Rap1(Q63E)-expressing HeLa cells in mitosis are shown that express the indicated CD8 chimera. Uptake was performed as previously described (*Fielding et al., 2012*). Internalised CD8 is shown in green, and surface CD8 is shown in red, and DNA in blue. Scale bars, 10 µm.

changes in mitotic cells (*Raucher and Sheetz, 1999*; *Stewart et al., 2011a*). In interphase cells, CME can overcome such increases in membrane tension by recruiting the actin cytoskeleton to assist in force production (*Aghamohammadzadeh and Ayscough, 2009*; *Boulant et al., 2011*). However, in mitotic cells, the cytoskeleton is engaged in the formation of a rounded actin cortex and is thus unavailable to assist the CME machinery. This lack of actin availability represents a second layer of inhibition. Local nucleation of actin could therefore provide an override mechanism for receptor internalisation in mitotic cells. Recently, F-actin-dependent internalisation of the EGF receptor has been described (*Vehlow et al., 2013*) and this may explain how the EGF receptor can be internalised in mitotic cells albeit with delayed kinetics (*Liu et al., 2011*). An interesting question for the future is how the CME machinery senses increased membrane tension in order to engage the actin cytoskeleton.

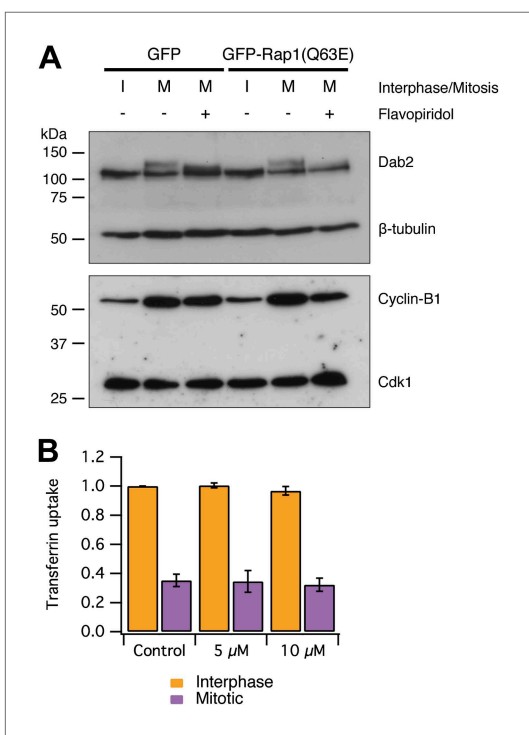

**Figure 6**. Mitotic phosphorylation does not explain mitotic shutdown of CME. (**A**) Western blot to show Dab2 phosphorylation and inhibition by the Cdk1 inhibitor flavopiridol (5 µM, 10 min). Interphase or mitotic cell lysates prepared from cells expressing GFP or GFP-Rap(Q63E) to restart CME in mitotic cells. Blots were probed for Dab2 and β-tubulin as a loading control, or Cyclin-B1 and Cdk1. (**B**) Bar chart to show the average transferrin uptake in interphase (orange) or mitotic (purple) populations of cells by flow cytometry. The mean ± SEM of three separate experiments are shown.

# Materials and methods

## Molecular biology

Plasmids for expression of mCherry-tagged mouse CTTN (ID27676) and mouse HIP1R tagged with tDimer-RFP (ID27700) were from Addgene. HIP1 tagged at the N-terminus with tdTomato was a kind gift from Uri Ashery (Tel Aviv University). GFP-tagged clathrin light chain a (GFP-LCa) was available from previous work (*Royle et al., 2005*). GFP-tagged Human Ect2 (isoform 2) and Rap1 plasmids, pRK5-Rap1-Q63E and pRK5-Rap1-S17A were kind gifts from Buzz Baum (MRC-LMCB, London, UK) (*Dao et al., 2009*). For Ect2 rescue experiments, siRNA resistant forms of GFP-Ect2 were made introducing silent mutations in the Ect2 siRNA target regions by site-directed mutagenesis. GFP-Rap1Q63E was constructed by digestion of pRK5-Rap1Q63E with EcoRI and ligation into pEGFP-C3. Rap1 constructs were expressed for 42 hr. The following siRNAs were used: CTTN (QIAGEN, UK Hs_CTTN_5 SI02661960 or Hs_CTTN_6 SI02662485), Ect2 (QIAGEN, Hs_ECT2_5 SI02643067 or Hs_ECT2_6 SI03049249) at 100 nM for ~66 hr or 42 hr, respectively.

## Cell culture and transfections

HeLa cells were cultured in DMEM containing 10% fetal bovine serum and 100 U/ml penicillin-streptomycin at 37°C, 5% $CO_2$. DNA transfections were performed with GeneJuice (MerckMillipore, UK) and siRNA transfections with Lipofectamine 2000 (Life Technologies, UK) according to the manufacturer's instructions.

For transferrin uptake analysis, HeLa cells were serum-starved in DMEM for 30 min at 37°C before Alexa-conjugated transferrin (50 µg/ml) was added

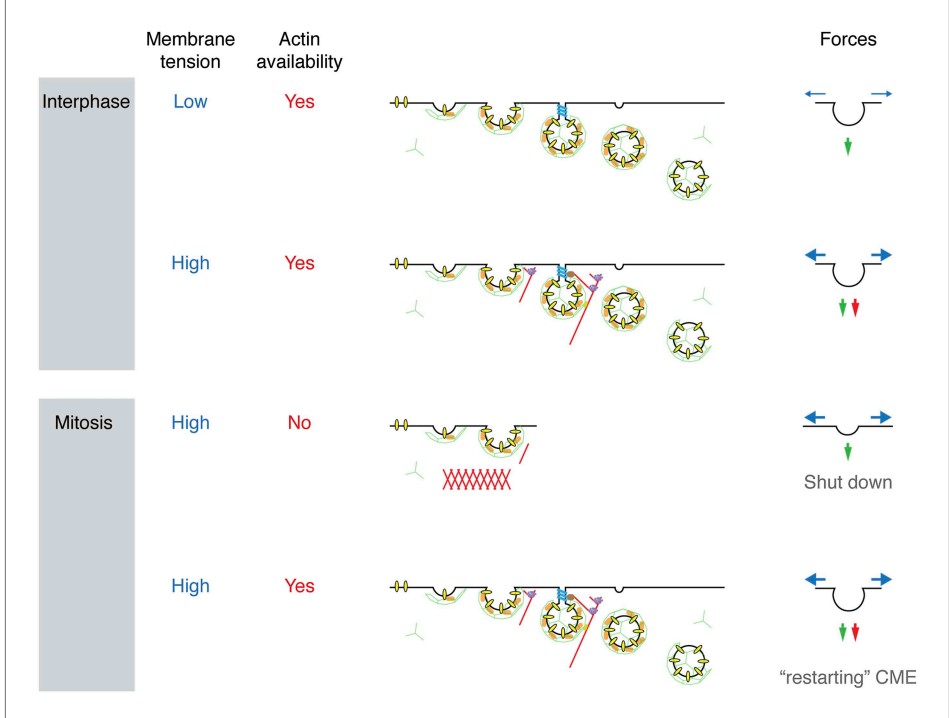

**Figure 7**. Model to show 'two layer' inhibition of CME during early mitosis. In interphase, CME is continually active and can deploy actin to assist endocytosis if the membrane tension is increased. The forces of CME, actin and membrane tension are represented as green, red and blue arrows, respectively. In mitosis, the membrane tension is increased and actin is required to assist CME, but it is also unavailable due to the formation of the actin cortex. Increasing actin availability, either by Ect2-depletion or expression of Rap1(Q63E), can restore CME in mitotic cells. The membrane tension remains high, but actin is available to assist CME by providing additional force. If the actin cytoskeleton is considered to be a component of the endocytic machinery then its unavailability can be regarded as being due to a 'moonlighting' function in the formation of a rounded cortex (**Royle, 2013**).

in serum-free DMEM. The cells were incubated for 7 min at 37°C to allow uptake and then placed on ice to halt endocytosis. Two acid-washes (100 mM Glycine, 150 mM NaCl in PBS, pH 3) were performed on ice for 5 min to remove surface-bound transferrin. Additionally, the cells were washed with PBS before and after each acid-wash to reduce surface binding of transferrin. Latrunculin B (1 µM) was added in serum-free media during the serum-starvation step of the transferrin uptake assay. For hypertonic sucrose treatment, sucrose (0.45 M) was added after 15 min of serum starvation.

## Immunostaining

HeLa cells were fixed with 3% paraformaldehyde/4% sucrose in PBS (wt/vol) for 15 min, permeabilised with 0.5% Triton X-100 in PBS (vol/vol) for 10 min, and then blocked for 30 min with 3% (wt/vol) BSA, 5% (vol/vol) goat serum in PBS. Primary antibodies (anti-CTTN, ab81208; Abcam, UK) were incubated for 2 hr in blocking solution. After washing, secondary antibodies (anti-rabbit Alexa Fluor568, 1:500) were added for 1 hr. Finally, after washing with PBS, the cells were mounted in ProLong Gold Antifade or Mowiol containing 10 µg/ml 4',6-diamidino-2-phenylindole (DAPI). Optionally, TRITC-phalloidin (Sigma-Aldrich, UK) was added at a 1:2000 dilution for 1 hr after the permeabilisation step.

CD8 live-labelling immunofluorescence experiments were carried out exactly as described previously (**Fielding et al., 2012**). For total CD8 immunostaining fixed cells were stained with anti-human CD8 Alexa488-conjugated antibody (1:200; MCA1226A488; Serotec, UK) with no secondary antibody.

To assess the distribution of G-actin and F-actin, the DNase I/phalloidin method was used (**Cramer et al., 2002**). Briefly, the cells were fixed in 4% formaldehyde (EM grade) with 0.32 M sucrose in cytoskeleton buffer (in mM: 10 Mes, 138 KCl, 3 MgCl$_2$, 4 EDTA, pH 6.1) for 20 min at room temperature. The cells were permeabilised and then blocked for 20 min at room temperature. Cover slips were incubated with 0.3 µM DNaseI-Alexa488 and 1:2000 Phalloidin-TRITC, for 20 min

at room temperature. The cover slips were washed three times in PBS with 0.1% Triton X-100 and then mounted as above.

## Clathrin-coated structure proteomics

The purification of fractions containing clathrin-coated structures was as described previously (*Borner et al., 2012*). HeLa cells were plated onto 245-mm square cell culture plates (~2 million cells/plate). Mitotic cells were synchronised by 24 hr in thymidine (2 mM), released for 3 hr and then 16 hr in nocodazole (100 ng/ml). This arrested ~80% of cells in mitosis and nine plates of mitotic cells were prepared vs seven plates for interphase. Mitotic cells were harvested 6 days after plating by shake-off collected by centrifugation, supernatant removed and cells were resuspended in Buffer A (0.1 M MES, pH 6.5, 0.2 mM EGTA, 0.5 mM MgCl$_2$, 0.02% NaN$_3$, 0.2 mM PMSF). Interphase cells were grown without synchronisation and were rinsed in Buffer A and scraped into Buffer A using a rubber policeman. The cells were lysed by 20 strokes in glass homogeniser (Kontes #22) and centrifuged at 152000 $g_{av}$ for 30 min at 4°C. Supernatant was removed to ice and 5 mg/ml RNAse A (Sigma-Aldrich) was added for 1 hr then centrifuged at 99,000 $g_{av}$ for 40 min at 4°C. Pellet was resuspended in 0.75 ml Buffer A and an equal volume of 12.5% Ficoll/12.5% sucrose was added and mixed in microfuge tubes and then centrifuged at 14,000 rpm for 30 min at 4°C in Eppendorf 5417R. Supernatant was removed, placed in thick-walled centrifuge tubes, mixed with 4 vol of Buffer A and clathrin-coated structures were pelleted by spinning at 121500 $g_{av}$ for 75 min at 4°C. Supernatant was removed and pellets resuspended in 20 µl Buffer A. The protein content was tested by BCA assay and equal amounts of interphase and mitotic samples (25–50 µg) were run on 4–12% gradient NuPAGE gels and stained with SimplyBlue SafeStain (Invitrogen). Each lane was cut into 32 slices and then digested with trypsin. The resulting tryptic peptide mixtures in 0.05% trifluoracetic acid were analyzed by nanoACQUITY UPLC system (Waters, UK), coupled to an LTQ Orbitrap XL mass spectrometer with a Proxeon nano-electrospray source. A 5 µl sample of the digest was injected into a BEH-C18 symmetry trapping column (Waters) in 0.1% formic acid at 15 µl/min before being resolved on a 25 cm × 75 µm BEH-C18 column (Waters), in a 1–62.5% acetonitrile gradient in 0.1% formic acid, with a flow rate of 400 nl/min. Full scan MS spectra (*m/z* 300–2000) were generated at 30,000 resolution, and the top five most intense ions were fragmented and subjected to MS/MS in the linear quadrapole ion trap. Ions were fragmented using collision-induced dissociation (collision energy 35%, 30 ms). All spectra were acquired using Xcalibur software (version 2.0.7). RAW files were analysed using MaxQuant version 1.2.7.4 and searched against the Human IPI database, v.3.77. The proteingroups.txt file from a combined MaxQuant analysis of all four experimental runs was used to calculate the interphase/mitotic ratio of LFQ values for each protein (*Waanders et al., 2009*). A Log2 transformation of each ratio was taken and used to construct a frequency histogram to which a Gaussian function was fitted to find the mean and variance of the entire dataset, and identify outliers (µ ± 2σ). MaxQuant detected the same outliers when each experiment was analysed individually. Excluded from the graph are proteins whose LFQ value for either interphase or mitosis was 0. This could be due to total loss of the protein or poor detection. The following proteins had large LFQ scores (>1 × 10$^6$) in one sample but 0 in the other and may therefore represent genuine CCS proteins that are only present in interphase (EPN1, HEXB, CTSC, NUMB) or in mitosis (CFL2, CLTCL1).

## Western blotting

HeLa cells transfected to express GFP or GFP-Rap1Q63E were synchronised with 100 ng/ml nocodazole for 16 hr at 37°C. Flavopiridol (5 µM) was added for 10 min immediately prior to lysis in RIPA buffer supplemented with PMSF (0.2 mM), NaF (30 mM), and okadaic acid (100 nM). Lysates were analysed by SDS-PAGE and western blotting. Commercially available antibodies were used to Dab2 (sc-13982; Santa Cruz/Insight Biotech, UK), cyclinB1 (554179; BD Biosciences, UK), cdk1 (ab18; Abcam), β-tubulin (ab6046; Abcam), Ect2 (SC-1005; Santa Cruz).

## Microscopy

Confocal imaging was done using a Leica confocal microscope SP2 with a 63X (1.4 NA) oil-immersion objective, as described previously (*Fielding et al., 2012*); or using a Perkin–Elmer UltraView spinning disk confocal microscope system using 405, 488 and 561 nm lasers and two Orca-R2 cameras (Hamamatsu) under the control of Volocity software (Perkin–Elmer). Epifluorescence images were taken using a Nikon-Ti Eclipse microscope with a 60X (1.4 NA) oil-immersion objective, DS-Qi1Mc camera and standard filter sets for visualisation of DAPI, GFP and Alexa568.

## Optical trapping

HeLa cells (400 µl at $5 \times 10^5$ cells/ml) were plated onto 22 × 50 mm glass cover slips (thickness 1.5). Each cover slip was suspended on blu tac pedestals inside a 10-cm petri dish alongside a 35-mm petri dish with no lid that contained 3 ml DMEM to humidify the larger dish. After 6 hr, 800 µl full DMEM was added to the cells and incubated overnight. This ensured good attachment of cells and a clean, dry surface on the underside of the glass.

Polybead carboxylate 1 µm microspheres (Polysciences/Park Scientific, UK) were coated with con-concanavalin A using a Polylink protein coupling kit. This gave 20 µg of protein per 1 mg of microparticle. For trapping experiments, the bead solution was diluted (1:20) in full DMEM. A microchamber was created by sealing a small cover glass over the slide on which the cells were grown using high vacuum grease (Apiezon). Bead suspension (10 µl) was introduced by capillary action and the microchamber was mounted onto a custom-made optical trap microscope, described elsewhere (*Carter and Cross, 2005*). The stiffness of the optical trap was calibrated for each experiment, typically ~0.3 pN/nm. A single bead was captured and held on the cell surface at a distance of 4–8 µm from the substrate, for several seconds. The bead was then pulled away, forming a membrane tether (>10 µm length), by moving the microscope stage under computer control at a constant velocity. The displacement of the bead in the optical trap (and thus tether force) was measured using a quadrant detector at 5–20 kHz. Once the displacement was established, the cell was moved back towards the bead faster than the tether recovery rate to provide a zero force reference (see *Figure 1C*). The cell was then moved back away and tether force was re-established. Tension values were taken from a 1 s averaged before stage manipulation. All measurements were taken less than 30 s after tether formation, because actin polymerisation into the tether introduced inconsistent measurements in older tethers. Measurements from multiple tethers pulled from the same cell indicated high reproducibility and showed that a single force measurement was representative of the membrane tension of that cell. Occasionally, double tethers visible in DIC were pulled by a single bead and these gave a twofold increase in tether force from the same cell. Data from double tethers were not included here.

## Flow cytometry

As previously described (*Fielding et al., 2012*), mitotic cells were synchronised with 100 ng/ml nocodazole for 18 hr and harvested by shake-off. Interphase cells were growing asynchronously and harvested by trypsinisation. The cells were spun down at 1000 rpm for 5 min and resuspended in serum free DMEM, and this step was repeated. For the mitotic samples, 100 ng/ml nocodazole was added to prevent progression through mitosis. After 20 min of incubation at 37°C, flavopiridol was added for a further 10 min at 37°C. The cells were moved to ice for 2 min then Alexa488-conjugated transferrin (50 µg/ml dilution) added on ice for 5 min. Uptake was allowed for 7 min at 37°C. Then, the cells were pelleted, resuspended in acid wash buffer and incubated on ice for 5 min. After a second acid wash, the cells were pelleted and resuspended in 1% (wt/vol) BSA/PBS. Finally, the samples were analysed by flow cytometry (FACSCalibur; Becton Dickinson). The forward scatter channel (FSC) was gated at 200 in order to exclude all particles smaller than cells from analysis. The side scatter channel (SSC) was adjusted to 320 V and the fluorescent channel (FL1) to 385 V. The geometric mean of FL1 was used for the graphs.

## Data analysis

Transferrin uptake quantification was performed in ImageJ using confocal images, as previously described (*Fielding et al., 2012*). For quantification of G-actin and F-actin fluorescence, a 24 pixel wide ROI was created at the cell border using a confocal z-section taken at the cell equator. A second ROI was created by placing a 50 × 50 ROI in the cytoplasm. The background-subtracted values were used to calculate the mean G-actin signal in the cytoplasm and a ratio of cortex/cytoplasm ratio for F-actin. All analysis, plots and statistical testing (see legends for details) were done in IgorPro 6.32A (WaveMetrics). Figures were assembled in Adobe Photoshop and Illustrator CS5.1.

## Acknowledgements

We thank Rachel Jones for technical help and Georg Borner and Scottie Robinson for help and advice on clathrin-coated vesicle proteomics. We also thank Uri Ashery and Helen Matthews for Rap1 plasmids. We are grateful to Corinne Smith for critical reading an early version of this manuscript and to members of the Royle lab for useful discussion. SJR is a Senior Cancer Research Fellow for Cancer Research UK.

# Additional information

## Funding

| Funder | Grant reference number | Author |
|---|---|---|
| Biotechnology and Biological Sciences Research Council | BB/H015582/1 | Stephen J Royle |
| Cancer Research UK | C25425/A15182 | Stephen J Royle |

The funder had no role in study design, data collection and interpretation, or the decision to submit the work for publication.

## Author contributions

SK, Acquisition of data, Analysis and interpretation of data, Drafting or revising the article; ABF, GG, NJC, Acquisition of data, Analysis and interpretation of data; SJR, Conception and design, Analysis and interpretation of data, Drafting or revising the article

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
