## [Decision Letter]

[Editors’ note: this article was originally rejected after discussions between the reviewers, but the authors were invited to resubmit after an appeal against the decision.]

Thank you for choosing to send your work entitled “An unmet actin requirement explains the mitotic inhibition of clathrin-mediated endocytosis” for consideration at *eLife*. Your full submission has been evaluated by a Senior editor and 3 peer reviewers, one of whom, Pascale Cossart, is a member of our Board of Reviewing Editors, and the decision was reached after discussions between the reviewers.

The manuscript addresses an important question and provides robust data suggesting that actin participates in cortical stiffness and is unavailable for CME during mitosis. However, this stiffness is not measured and direct measurement of actin availability is also not made. These experiments may take time. In the spirit of our policy to ask for revisions only when these are straightforward, we regret to inform you that your work will not be considered further for publication at this point.

*Reviewer*
*#1:*

In this manuscript the authors address the long standing issue that endocytosis is less efficient during mitosis compared to interphase. They identify cortactin as the protein whose levels are the more dramatically changed, in mitotic versus interphase cellular fractions considered as enriched in clathrin coated structures (CCSs).

This leads the authors to the hypothesis that the cytoskeleton dynamics might be a key issue. Using two approaches they show that restoring actin dynamics leads to an increase in transferrin uptake in mitotic cells. In addition, they show that dab2 phosphorylation in mitosis does not regulate the endocytosis pathway. This is a convincing manuscript but controls are lacking such as rescue experiments of RNAi treated cells. The literature on the role of actin in endocytosis is not correctly cited.

*Reviewer*
*#2:*

The paper by Fielding, Gassner and Royle presents evidence that clathrin mediated endocytosis (CME) in mitotic cells is inhibited due to sequestration of actin at the actin cortex. The authors propose that this, combined with increased membrane tension, effectively blocks CME of transferrin receptor (TfnR).

This is a short and clearly presented paper with a straightforward story. I have only a few minor questions.

1) Can the authors discount indirect effects and / or alternative mechanisms by which Rap1/Ect2 expression manipulations affect CME?

2) Hip1 and Hip1R are respectively enriched and depleted in clathrin coated structures (CCS) from mitotic cells. Can the authors comment on the difference in distribution between these two similar proteins?

3) Can the authors test whether hypo-osmotic conditions (relieving the presumed increase in mitotic cell surface tension) ‘re-starts’ CME?

4) Strikingly, the most enriched protein in mitotic CCS is NSF. Can the authors comment?

5) Flavopiridol (the Cdk1 inhibitor) did seem to reduce the phosphorylation of Dab2 a little but since this adaptor is not implicated in TfR endocytosis I’m not sure how this is relevant. There are a lot of different kinases implicated either directly or indirectly in controlling CME, so pinning everything on Cdk1 seems a bit weak. Then again, perhaps control could be exerted by other kinases downstream of Cdk1? Perhaps this would just be better in supplementary information?

*Reviewer*
*#3:*

Here the authors seek to understand the mechanism by which clathrin-mediated endocytosis (CME) is downregulated during mitosis. In particular, they seek to distinguish between two alternative mechanisms of downregulation -phosphorylation and inhibition of CME proteins, or increased membrane tension involving the actin cytoskeleton. Their evidence suggests that regulation by phosphorylation does not play a primary role. Instead they find that indirectly perturbing cortical actin enables CME during mitosis, suggesting that modulating cortical tension or stiffness is important for downregulating mitotic CME. Based on this, they propose a model in which mitotic CME is inhibited because actin participates in cortical stiffness and is unavailable for CME.

On the positive side, this manuscript addresses a very interesting topic, and the findings are suggestive. Nevertheless, the major conclusions are not well supported by the data provided, as outlined below:

1) How accurate is the mass-spec quantification of protein abundance in interphase versus mitosis (Figure 1)? The authors are careful to carry out four independent experiments, and identify proteins that are either enriched or depleted in mitotic cells relative to interphase cells. However, they should offer a better explanation of the methods for quantifying relative protein abundance.

2) The authors propose the general idea that cortical actin negatively regulates mitotic CME. The data to support this are provided in Figures 2 and 3. Figure 2 shows that expression of a constitutively activated Rap1 (which supposedly disrupts cortical actin and prevents mitotic rounding) activates CME in mitosis, and this activation is prevented by depleting cortactin. The conclusion is that “actin is required, but unavailable for, CME in mitotic cells”. However, there is no direct test of actin “availability” to support this conclusion, and it is unclear what the authors mean by the word “availability”. Are they suggesting that the ratio of G-actin to F-actin is altered in mitotic versus interphase cells? Or, are they suggesting that actin filaments are somehow unavailable based on their localization to the cortex? They should more carefully define their hypothesis, and provide more direct tests of their ideas.

Along these same lines, Figure 3 shows that depletion of Ect2 (a Rho GEF that affects cortical actin) also activates CME in mitosis, and that this effect is suppressed by adding the actin sequestering drug latrunculin B. Again, the conclusion is that “an unmet requirement for actin in CME causes shutdown.” Although the activated CME is clearly actin dependent, it is unclear from these results whether enabling actin function in mitosis, or making it more “available”, would be sufficient to restart CME. One possible test would be to express constitutively active ADF/cofilin, which might render actin more “available”, and test whether this enables mitotic CME. In any case, additional evidence is needed to support the conclusion.

3) To test for phosphoregulation of CME during mitosis, the authors briefly treat cells with the Cdk1 inhibitor flavopiridol, which inhibits the mitotic phosphorylation of the CME protein Dab2 (Figure 4). Flavopiridol does not affect mitotic CME, leading the authors to conclude that “the direct mitotic phosphorylation of endocytic proteins cannot account for mitotic shut down of CME”. However, this one experiment is not sufficient to support this conclusion. The time period of inhibition (10 min) may not be enough to reduce mitotic phosphorylation of all CME proteins. Also, additional inhibitors or RNAi should be employed to bolster the conclusions. It would be surprising if mitotic phosphorylation plays no role in inhibition of CME, as it is the major mechanism for regulating the mitotic transition.

Minor comments:

1) Figure 2: The authors should test the effect of cortactin RNAi on CME in cells not expressing Rap1(Q63E).

2) Figure 2: A Western blot showing cortactin depletion should also be shown, and multiple cortactin siRNAs should be employed as controls, as is done for Ect2.

---

## [Author Response]

Reviewer #1:

*In this manuscript the authors address the long standing issue that endocytosis is less efficient during mitosis compared to interphase. They identify cortactin as the protein whose levels are the more dramatically changed, in mitotic versus interphase cellular fractions considered as enriched in clathrin coated structures (CCSs)*.

*This leads the authors to the hypothesis that the cytoskeleton dynamics might be a key issue. Using two approaches they show that restoring actin dynamics leads to an increase in transferrin uptake in mitotic cells. In addition, they show that dab2 phosphorylation in mitosis does not regulate the endocytosis pathway*.

We thank the reviewer for their time and constructive comments on our manuscript. We were very pleased that they recognised the importance of the question that we are addressing and that they found our manuscript convincing.

*This is a convincing manuscript but controls are lacking such as rescue experiments of RNAi treated cells*.

There are two RNAi experiments in the paper, depletion of Ect2 and depletion of CTTN, and we have now performed rescue experiments for both. For the Ect2 knockdown experiment, we have tested for transferrin uptake in cells expressing GFP or GFP-Ect2 (siRNA resistant), cells were treated with control or Ect2 siRNA. We see restarting of mitosis only when endogenous Ect2 is depleted and no exogenous Ect2 is expressed. This is now presented as Figure 3—figure supplement 1 and shows that restarting of CME in Ect2-depleted cells is due to loss of Ect2 protein.

For the CTTN knockdown in Rap1(Q63E)-expressing cells, which prevents restoration of CME in mitotic cells, we have now included a second CTTN siRNA that shows the same effect. We have also performed a rescue experiment that demonstrates that the loss of CTTN is responsible for preventing restoration of CME in mitotic cells expressing Rap1(Q63E). The only conditions in which we see mitotic CME is when CTTN is present when Rap1(Q63E) is expressed. The controls for this experiment, where mCherry-cortactin is expressed in control mitotic cells also allow us to make the point that it is not simply loss of CTTN during mitosis that causes mitotic shutdown of CME. This was one possible interpretation of our proteomics experiments.

*The literature on the role of actin in endocytosis is not correctly cited*.

We are not sure to which literature the reviewer is referring. One omission was for us not to acknowledge that in mammalian cells, actin is essential for pathogen entry. We have now added a sentence to rectify this. In light of the comment from Reviewer #2, we have reworded the sentence on whether actin has an obligatory role in mammalian CME.

Reviewer #2:

*The paper by Fielding, Gassner and Royle presents evidence that clathrin mediated endocytosis (CME) in mitotic cells is inhibited due to sequestration of actin at the actin cortex. The authors propose that this, combined with increased membrane tension, effectively blocks CME of transferrin receptor (TfnR)*.

*This is a short and clearly presented paper with a straightforward story. I have only a few minor questions*.

We thank the reviewer for their time and constructive comments on our manuscript. We were very pleased that they had only minor comments.

*1) Can the authors discount indirect effects and / or alternative mechanisms by which Rap1/Ect2 expression manipulations affect CME*?

We have added several experiments that demonstrate that restoration of CME in mitotic cells by either Rap1(Q63E) expression or Ect2 depletion is due to making actin available to assist the CME machinery to overcome increased membrane tension. In brief, the manuscript now shows:

1. Membrane tension is elevated in mitotic cells and stays elevated after Rap1(Q63E) expression or Ect2 depletion

2. Rap1(Q63E) expression or Ect2 depletion make more actin available (G-actin increase) by preventing F-actin to be remodelled at the cortex (F-actin ratio decrease)

3. The restored CME is actin-dependent. It can be blocked by latrunculin B treatment (for Ect2 depletion) or by cortactin RNAi (for Q63E expression).

4. Restarting of CME in Ect2-depleted mitotic cells can be blocked by expression of siRNA-resistant forms of GFP-Ect2. Two Ect2 siRNAs are used and rescued, showing that the phenomenon is due to loss of Ect2.

5. Cancelling restarted CME in Rap1(Q63E)-expressing mitotic cells by depletion of cortactin can itself be blocked by expression of siRNA-resistant mCherry-cortactin. This shows that the specificity of cortactin depletion is due to loss of cortactin protein and is not a result of an off-target effect.

*2) Hip1 and Hip1R are respectively enriched and depleted in clathrin coated structures (CCS) from mitotic cells. Can the authors comment on the difference in distribution between these two similar proteins*?

We have now localised the distribution of HIP1R, HIP1, and CTTN together with clathrin in interphase and mitotic cells to verify these results from the proteomic survey. This is now shown in the main proteomics figure (now Figure 2). We find that all three co-localise with some clathrin spots in interphase. For HIP1 and CTTN, the proteins become diffusely localised in mitosis, i.e. the association with clathrin is reduced. We also found a similar pattern with antibodies to the endogenous proteins (not shown) and so these findings agree with the comparative proteomics.

We do not know why HIP1R and HIP1 are differentially regulated in this way. This is an important question for the future. We have made a series of cortactin mutants that block and mimic mitotic phosphorylation, none of which did anything noticeable to the distribution of cortactin in mitosis. We feel that understanding the mislocalisation of HIP1 and cortactin is key to this difference. We will investigate this in the future but feel that this is beyond the scope of this study.

*3) Can the authors test whether hypo-osmotic conditions (relieving the presumed increase in mitotic cell surface tension) ‘re-starts’ CME*?

We have attempted this manipulation along with many others, summarised briefly below. The short answer is “no, it doesn’t”, probably due to homeostasis.

We have carried out a serial dilution toward hypo-osmotic conditions (see Figure 8 below). This manipulation eventually inhibits CME in interphase cells, but CME in mitotic cells in unaffected. To decrease membrane tension hyper-osmotic conditions are required, to shrink the cell. This condition (e.g., 0.45M sucrose) is known to inhibit CME by making clathrin form microcages. We tried to dial this back using different salt concentrations and got good inhibition in the interphase cells and no increase in mitotic cells.Author response image 1.Osmolarity change and CME.Transferrin uptake in HeLa cells treated as indicated by diluting the media by the indicated amount of water. Geo Mean from a population of >1 x 10^4 cells is shown.

I think the reason this doesn’t work is that the cell responds really fast to changes in osmolarity to re-establish membrane tension. In Figure 3 of Stewart et al. 2011 (PMID 21196934) you can see that cells rebound in 2 min after a change in osmolarity. This is too fast for our endocytic measurements.

We have also tried to add the agents originally used by Raucher & Sheetz (1999, PMID 9971744) to reduce membrane tension. These are coarse reagents like DMSO, ethanol, and deoxycholate (!). We found that they permeabilise the plasma membrane allowing non-specific entry of fluorescent transferrin.

*4) Strikingly, the most enriched protein in mitotic CCS is NSF. Can the authors comment*?

We have used the proteomics data to point us in the right direction of actin. There are quite a few interesting things in the dataset and we are currently following some of these up. We have added a sentence to draw attention to NSF as the protein that accumulates most on mitotic CCSs. We have also added details of several proteins that were off-the-dial and not plotted (see response to Reviewer #3 comment 1). Uncoating is known to be inhibited in mitotic cells and our best guess is that this is why we see some proteins accumulating in the mitotic sample.

*5) Flavopiridol (the Cdk1 inhibitor) did seem to reduce the phosphorylation of Dab2 a little but since this adaptor is not implicated in TfR endocytosis I’m not sure how this is relevant. There are a lot of different kinases implicated either directly or indirectly in controlling CME, so pinning everything on Cdk1 seems a bit weak. Then again, perhaps control could be exerted by other kinases downstream of Cdk1? Perhaps this would just be better in supplementary information*?

We thank the reviewer for feedback on this point. We have now tested whether endocytosis of the three main endocytic motifs is restored by expression of Rap1(Q63E). To do this, we used CD8 chimeric reporters with YXXΦ, [DE]XXXL[LI], and FXNPXY. We found that all of the reporters are restarted. Importantly this includes an FXNPXY reporter that is the type of motif recognised by Dab2 (34). This makes a link to the experiments showing that Dab2 remains phosphorylated in mitotic cells expressing Rap1(Q63E). As FXNPXY motifs can be internalised in these cells then it argues that the phosphorylation (of Dab2 at least) cannot be inhibitory. We think these experiments are important and they remain in the main paper (new Figure 6).

We accept the criticism that pinning everything on Cdk1 (albeit the master kinase in mitosis) and only looking at one adaptor is not comprehensive. We have now tempered this section by noting that mitotic phosphorylation could modulate the shut down process but that it cannot be solely responsible. We feel this is the most parsimonious explanation.

Reviewer #3:

*Here the authors seek to understand the mechanism by which clathrin-mediated endocytosis (CME) is downregulated during mitosis. In particular, they seek to distinguish between two alternative mechanisms of downregulation -phosphorylation and inhibition of CME proteins, or increased membrane tension involving the actin cytoskeleton. Their evidence suggests that regulation by phosphorylation does not play a primary role. Instead they find that indirectly perturbing cortical actin enables CME during mitosis, suggesting that modulating cortical tension or stiffness is important for downregulating mitotic CME. Based on this, they propose a model in which mitotic CME is inhibited because actin participates in cortical stiffness and is unavailable for CME*.

*On the positive side, this manuscript addresses a very interesting topic, and the findings are suggestive. Nevertheless, the major conclusions are not well supported by the data provided, as outlined below*:

We thank the reviewer for their time and constructive comments on our manuscript. We were very pleased that they recognised the question that we are addressing is very interesting. We have undertaken substantial revisions that better support our major conclusions.

*1) How accurate is the mass-spec quantification of protein abundance in interphase versus mitosis (*Figure 1*)? The authors are careful to carry out four independent experiments, and identify proteins that are either enriched or depleted in mitotic cells relative to interphase cells. However, they should offer a better explanation of the methods for quantifying relative protein abundance*.

We think that the mass-spec quantitation is certainly accurate enough to show us which proteins are altered and in which direction. Even with the variability of the CCS proteome in this preparation (we were especially stringent in using 2 standard deviations as the limit for variability). For absolute quantification, to confirm how many copies of a given protein are present per clathrin triskelion, then a more detailed analysis would be required ([5] PMID 22472443). The absolute quantitation is interesting, but it is not the focus here.

To verify our proteomic analysis, we have now added localisation experiments to show that the relative loss of HIP1 and CTTN in CCSs in mitosis is a real phenomenon that can be visualised in cells. Our proteomic analysis also found Dab2 and PICALM as depleted or enriched on mitotic CCSs, respectively, which had been reported previously. We were pleased that the reviewer recognised our efforts to show the reproducibility of these changes using four independent (completely independent) proteomic experiments.

For a better explanation of the methods, we have now expanded the methods section to better describe how this part of the work was done. One limitation to our approach is that any proteins that are present in one sample, but not detected in the other sample, are not included on the bar chart in Figure 2, because log2 of x/0 or 0/x is ∞ or −∞, respectively. We excluded those from the plot as this could be a detection issue rather than an absolute loss of the protein from one sample. A note about this has now been added.

*2) The authors propose the general idea that cortical actin negatively regulates mitotic CME. The data to support this are provided in*
Figures 2 and 3*.*
Figure 2
*shows that expression of a constitutively activated Rap1 (which supposedly disrupts cortical actin and prevents mitotic rounding) activates CME in mitosis, and this activation is prevented by depleting cortactin. The conclusion is that “actin is required, but unavailable for, CME in mitotic cells”. However, there is no direct test of actin “availability” to support this conclusion, and it is unclear what the authors mean by the word “availability”. Are they suggesting that the ratio of G-actin to F-actin is altered in mitotic versus interphase cells? Or, are they suggesting that actin filaments are somehow unavailable based on their localization to the cortex? They should more carefully define their hypothesis, and provide more direct tests of their ideas*.

*Along these same lines,*
Figure 3
*shows that depletion of Ect2 (a Rho GEF that affects cortical actin) also activates CME in mitosis, and that this effect is suppressed by adding the actin sequestering drug latrunculin B. Again, the conclusion is that “an unmet requirement for actin in CME causes shutdown.” Although the activated CME is clearly actin dependent, it is unclear from these results whether enabling actin function in mitosis, or making it more “available”, would be sufficient to restart CME. One possible test would be to express constitutively active ADF/cofilin, which might render actin more “available”, and test whether this enables mitotic CME. In any case, additional evidence is needed to support the conclusion*.

Our model for mitotic shutdown of CME (Figure 7) is that actin is required to assist the CME machinery to overcome the increase in membrane tension in mitotic cells. Because actin is remodelled into the actomyosin cortex in mitotic cells we propose that it is unavailable to assist CME.

In the previous version of the manuscript we had shown that we could restart CME in mitotic cells and that this endocytosis was uniquely actin-dependent. We proposed that this was due to “freeing-up” actin by inhibiting cortex formation; we also assumed that the membrane tension remained high following these manipulations. We have now undertaken significant experimental work that directly tests our ideas.

1. We have measured F-actin and G-actin by light microscopy in cells with Ect2 depletion or with Rap1(Q63E) expression. This shows that the F-actin at the cortex is reduced relative the cytoplasmic level. Also, the G-actin levels in the cytoplasm are increased (a small but significant increase of 10-20%).

2. We have measured membrane tension using laser tweezers and confirmed that tension remains high in mitotic cells that are either depleted of Ect2 or are expressing Rap1(Q63E).

3. Just to be clear, we are not proposing “the general idea that cortical actin negatively regulates mitotic CME” if the reviewer thought that we meant that cortical actin physically gets in the way of CME. We have now added a sentence to the discussion of the latrunculin B experiments to make this point more clearly. Latrunculin B treatment of mitotic cells does not cause restarting of CME that would be expected if the cortical actin directly inhibits CME.

*3) To test for phosphoregulation of CME during mitosis, the authors briefly treat cells with the Cdk1 inhibitor flavopiridol, which inhibits the mitotic phosphorylation of the CME protein Dab2 (*Figure 4*). Flavopiridol does not affect mitotic CME, leading the authors to conclude that “the direct mitotic phosphorylation of endocytic proteins cannot account for mitotic shut down of CME”. However, this one experiment is not sufficient to support this conclusion. The time period of inhibition (10 min) may not be enough to reduce mitotic phosphorylation of all CME proteins. Also, additional inhibitors or RNAi should be employed to bolster the conclusions. It would be surprising if mitotic phosphorylation plays no role in inhibition of CME, as it is the major mechanism for regulating the mitotic transition*.

We thank the reviewer for their feedback on this point. We were obviously not clear enough about our interpretation of these experiments and we have addressed this in the revised version.

The time point of inhibition with Flavopiridol was selected so that cells do not reverse out of mitosis back into G2 (Potapova et al., 2006 PMID 16612388). It is possible that phosphorylation of all CME proteins may not be reduced in this time, but this is not the sole set of experiments on which we pin our conclusion.

We have now tested whether endocytosis of the three main endocytic motifs is restored by expression of Rap1(Q63E). To do this, we used CD8 chimeric reporters with YXXΦ, [DE]XXXL[LI], and FXNPXY. We found that endocytosis of all these reports is restarted. Importantly, this includes an FXNPXY reporter that is the type of motif recognised by Dab2 (34). This makes a link to our experiments, which show that Dab2 remains phosphorylated in mitotic cells expressing Rap1(Q63E). As FXNPXY motifs can be internalised in these cells then it argues that the phosphorylation (of Dab2 at least) cannot be inhibitory.

We completely agree that phosphorylation is involved in the shutdown at some level because phosphorylation governs mitotic entry and so it is in overall control of the phenomenon. We weren’t clear about that and now we state it unequivocally. What we are questioning is the hypothesis that mitotic phosphorylation of individual CME proteins decreases their activity to cause the shutdown. Given the comments of this reviewer and Reviewer #2, we have now tempered our discussion of these results by noting that mitotic phosphorylation could modulate the shut down process but that it cannot be solely responsible. We feel this is the most parsimonious explanation.

*Minor comments*:

*1)*
Figure 2*: The authors should test the effect of cortactin RNAi on CME in cells not expressing Rap1(Q63E)*.

This has now been done (see Figure 4–figure supplement 2). There is no change in CME. Interphase cortactin-depleted cells have CME, and mitotic cortactin-depleted cells do not.

*2)*
Figure 2*: A Western blot showing cortactin depletion should also be shown, and multiple cortactin siRNAs should be employed as controls, as is done for Ect2*.

There are two RNAi experiments in the paper, depletion of Ect2 and depletion of CTTN, and we have performed rescue experiments for both. For the Ect2 knockdown experiment, we have tested for transferrin uptake in cells expressing GFP or GFP-Ect2 (siRNA resistant). These cells were treated with control or Ect2 siRNA. We see restarting of mitosis only when endogenous Ect2 is depleted and no exogenous Ect2 is expressed. This is now presented as Figure 3—figure supplement 1 and shows that restarting of CME in Ect2-depleted cells is due to loss of Ect2 protein.

For the CTTN knockdown in Rap1(Q63E)-expressing cells, which prevents restoration of CME in mitotic cells, we have now included a second CTTN siRNA that shows the same effect. We have also performed a rescue experiment that demonstrates that the loss of CTTN is responsible for preventing restoration of CME in mitotic cells expressing Rap1(Q63E). The only conditions in which we see mitotic CME is when CTTN is present when Rap1(Q63E) is expressed. This is now presented in Figure 4–figure supplement 2.

We show in Figure 4 the immunofluorescent staining of cortactin in the same cells in which we have measured transferrin uptake. This allows us to confirm that the knockdown occurs in the Rap1(Q63E)-expressing cells, which is not possible with western blotting.